# Infection control in the home: a qualitative study exploring perceptions and experiences of adhering to protective behaviours in the home during the COVID-19 pandemic

Katherine Morton ,[1] Lauren Towler ,[1] Julia Groot,[2] Sascha Miller ,[1] Ben Ainsworth ,[2,3] James Denison-Day ,[1] Cathy Rice ,[4] Jennifer Bostock ,[4] Merlin Willcox ,[5] Paul Little ,[5] Lucy Yardley [1,6]

KM and LT are joint first authors.

For numbered affiliations see end of article.

**Correspondence to**
Lauren Towler;
lbt1g14@soton.ac.uk

## ABSTRACT

**Objectives** We sought to explore people's experiences and perceptions of implementing infection control behaviours in the home during the COVID-19 pandemic, guided by an online behavioural intervention.

**Design** Inductive qualitative study.

**Setting** UK public during the COVID-19 pandemic.

**Participants** Thirteen people took part in telephone interviews, and 124 completed a qualitative open-text survey. All were recruited from the public. Most survey participants were aged over 60 years, while interview participants were more distributed in age. Most reported being at increased risk from COVID-19, and were white British.

**Intervention** Online behavioural intervention to support infection control behaviours in the home during the COVID-19 pandemic.

**Data collection** Telephone think-aloud interviews and qualitative survey data.

**Data analysis** The think-aloud interview data and qualitative survey data were analysed independently using inductive thematic analysis. The findings were subsequently triangulated.

**Results** Thematic analysis of the telephone interviews generated seven themes: *perceived risk; belief in the effectiveness of protective behaviours; acceptability of distancing and isolation; having capacity to perform the behaviours; habit forming reduces effort; having the confidence to perform the behaviours; and social norms affect motivation to engage in the behaviours.* The themes identified from the survey data mapped well onto the interview analysis. Isolating and social distancing at home were less acceptable than cleaning and handwashing, influenced by the need for intimacy with household members. This was especially true in the absence of symptoms and when perceived risk was low. People felt more empowered when they understood that even small changes, such as spending *some* time apart, were worthwhile to reduce exposure and lessen viral load.

**Conclusions** The current study provided valuable insight into the acceptability and feasibility of protective behaviours, and how public health guidance could be

### Strengths and limitations of this study

► To our knowledge, this is the first paper to qualitatively explore attitudes towards and experiences of performing protective behaviours within the home to prevent within-household transmission, which has been shown to be a key risk.

► Think-aloud interview data were triangulated with data from 124 qualitative survey respondents, and affinity between the two data sources was high.

► Transferability of the results is potentially limited due to the rapidly shifting nature of the pandemic, and limited representation of participants from minority ethnic groups.

► In addition, the qualitative survey had a low response rate which could limit transferability.

incorporated into a behaviour change intervention for the public during a pandemic.

## INTRODUCTION

Behavioural measures have been recommended to help control the spread of the COVID-19 virus, including handwashing, cleaning surfaces, mask-wearing, social isolation and social distancing.[1] However, evidence suggests that adherence to these behaviours varies widely in the UK and other affected countries, suggesting there may be challenges for people in implementing these behaviours in a real life setting.[2–4] Transmission of COVID-19 within the home is a key risk,[5 6] therefore understanding barriers to adhering to protective behaviours within the home could be particularly important.

Germ Defence is an infection control intervention which was initially developed using theoretical modelling and qualitative research to target seasonal colds and influenza, in line

with the person-based approach.[7] The intervention has been updated and optimised by the universities of Bristol, Bath and Southampton to help people protect themselves at home from COVID-19,[2 8] and its implementation into primary care is currently being trialled.[9] During the development of Germ Defence, the theory of planned behaviour (TPB) was applied to identify behavioural determinants on which to base the content.[10] Leventhal's common-sense model of health and illness was used to ensure the website content attended to common perceptions and constructions of illness and infection.[11] To increase users' perceived risk, the intervention is structured using protection motivation theory (PMT) by emphasising the personal and social health consequences of contracting COVID-19.[12] Evidence suggests that TPB and PMT concepts in particular explain behavioural responses during a pandemic.[13] Risk messages are followed by supportive coping messages explaining how users can reduce that risk by lowering their contact with the virus. The language used on the website is in line with the self-determination theory to increase users' motivation to carry out the behaviours.[14] Intervention content, design and structure were informed by qualitative think-aloud interviews with the general public.[15]

This study sought to explore experiences and perceptions of performing protective behaviours at home in order to identify possible barriers and facilitators, and develop an understanding of how these behaviours are influenced by perceptions. This forms part of the person-based approach to adaptation and optimisation of the Germ Defence intervention for COVID-19.[16]

## METHOD
### Participants
Inclusion criteria were those over the age of 18 years, able to access the Germ Defence website and able to give informed consent. Users of the Germ Defence website were invited to register their interest in taking part in research to optimise the website.

### Interviews
Seven interview participants were purposively sampled from the volunteers by factors such as age, gender, education level, risk status and experience of COVID-19 to maximise diversity.

However, after seven interviews we identified that these participants were mostly highly educated about infection control behaviours and highly motivated to adhere. As we wanted to understand barriers among people with lower levels of awareness and motivation, we recruited the remaining participants via social media and newsletters sent out by organisations and community groups to target people who had not already sought out the Germ Defence intervention (n=6). We stopped recruiting once we felt we had reached saturation and that no new barriers or facilitators were being identified.

### Survey
Users of Germ Defence who volunteered to participate in research but were not purposively sampled for an interview were invited to complete a short survey instead.

### Measures
#### Demographics
Potential participants were asked to complete an online survey to determine age, gender, experience of COVID-19, education, household size, postcode to inform Index of Multiple Deprivation, and ethnicity. Finally, contact information was collected to enable a researcher to invite the potential participant to interview or to complete the survey.

#### Interview topic guide
Think-aloud semistructured interviews[17 18] were conducted by three female interviewers (LT, KM and JG), in which the participants provided feedback on each page of the online intervention (https://www.germdefence.org/) to provide detailed insights into their perceptions of the content.[2] At the beginning of the interviews, participants were asked a series of questions pertaining to their general perceptions of COVID-19 and protecting themselves at home (eg, 'Can you tell me how you feel about the coronavirus at the moment?'). Then, the participants used the website and the researcher asked them what they thought of the content on each page. All interviewers were researchers within the field of health psychology. Prompts or follow-up questions typically pertained to attitudes towards the behavioural information and determinants of engagement and adherence. At the close of the interview, a series of general questions were asked about their overall views of the Germ Defence website.

#### Survey
The qualitative survey featured four open-text questions in addition to closed demographics questions. The survey aimed to gather participants' thoughts on the protective behaviours suggested on the website such as, 'How do you feel about following the suggestions on Germ Defence?' and 'What did you not like about the Germ Defence advice?'.

### Procedure
Those who accessed the Germ Defence website and completed at least one section saw a pop-up banner asking if they might be interested in taking part in research to help improve the website. If they indicated they wished to take part in research they were asked to complete the online demographic questions hosted by Qualtrics to inform purposive sampling. In addition, adverts inviting people to take part in a telephone interview about a website designed to help keep them and their household safe from coronavirus were posted on social media, with a link to the purposive sampling questions.

## Interviews

Participants were purposively selected by the research team and sent a link to the information sheet and consent form, which was completed online. Interviews were conducted by telephone, due to the pandemic. The audio recording began once consent was verbally reaffirmed. At the close of the interview, participants were thanked with an Amazon voucher. The interviews took place during a period of rapidly changing guidelines in the UK, from 8 June to 5 November 2020, most while the R-rate was relatively low, and restrictions were soon to be (or had already been) lifted.

## Survey

A total of 545 respondents were invited to complete the survey over three separate mail-outs: the first on 19 June 2020 (n=150); the second on 10 July 2020 (n=103); and the third on 24 July 2020 (n=292). The email contained a link to the survey, which began with a participant information sheet and consent form. For context, the first mail-out occurred during the first lockdown, which was lifted on 4 July 2020, but wearing face coverings inside shops only became compulsory on the date of the final mail-out; 24 July 2020.

## Patient and public involvement

As Germ Defence is available to the general public, patient and public involvement (PPI) was integral to its development. Two public contributors (CR and JB) on our stakeholder panel participated in weekly meetings which informed the optimisation of the intervention, and worked with us to identify potential issues in the behavioural messages of the intervention and update the intervention content in line with feedback. The conceptualisation, measures, recruitment strategy and dissemination of the current study was informed by open discussion with these members. For example, the public contributors reviewed the interview topic guide and assisted in identifying which organisations to target during the recruitment process. In particular, the public contributors provided considerable assistance in ensuring that the study materials and study invitations were easy to understand and free of jargon. Further detail on PPI in the development and optimisation of Germ Defence has been reported elsewhere.[16]

## Data analysis

### Interviews

Data were analysed using inductive thematic analysis to openly explore the barriers and facilitators that were important to people.[19 20] Due to the need for rapid analysis and dissemination of initial findings, the first set of transcripts was split between two researchers (n=6 transcripts analysed by KM and n=3 transcripts by LT). The researchers independently read their transcripts thoroughly to first familiarise themselves with the data. Data were then coded inductively by unit of meaning using NVivo, keeping the core aims of the study in mind (barriers and facilitators to, and perceptions of, infection control behaviours in the home). After the first nine interviews had been coded, the researchers met and compared their coding manuals, discussing each code and theme in detail and generating a final agreed coding manual to unite their coding. This involved revisiting the raw data to confirm shared and consistent understanding of how the codes and themes were being used. The coding manual was then used by LT to code the remaining four interviews, and where necessary new codes were added and existing codes were further refined, although these amendments were only minor. LT double-checked the earlier transcripts to ensure the revised coding manual was consistently applied across the data, and the researchers met again to confirm agreement on the final coding manual. Findings were shared with participants via a newsletter, and participants were invited to contact the research team if they had any feedback on the findings.

### Survey

Responses to the four open-text survey questions were coded inductively using thematic analysis, separately from the interview data analysis. The resulting categories were then mapped onto the themes generated from the interview data to assess their fit with these themes, whether any new themes or subthemes were present in the survey data, and to what extent the survey data provided further nuance to the existing themes. Inductive coding was deemed most appropriate, as the researchers intended to triangulate the two data sets for complementarity, rather than convergence, to ensure that any unique perspectives gathered from the survey data were attended to.

## RESULTS

Table 1 shows the demographic details of the 13 interview participants. The mean interview length was 79 min (range 60–104 min). Most participants lived with at least one other person, and seven participants felt that either they or a household member were at increased risk should they contract the virus.

A total of 124 website users completed the qualitative survey (n=545 invited, 23% response rate). Most participants were over 60 years old, reported being at increased risk from COVID-19, and white British. Table 2 shows the demographic details of the survey respondents.

The researchers generated seven key themes from the interview data related to perceived barriers and facilitators to engaging with infection control behaviours in the home. These were: *perceived risk; belief in the effectiveness of protective behaviours; acceptability of distancing and isolation; having capacity to perform the behaviours; habit forming reduces effort; confidence in how to perform the behaviours*; and *social norms affect motivation to engage in the behaviours*. See online supplemental file 1 for the coding manual. Extracts from the interview data are delineated by the abbreviation 'int'.

For the qualitative survey, most respondents felt positively about the protective behaviours recommended on the Germ Defence website. The themes identified from the survey data mapped well onto the interview analysis, with particularly strong congruence to *confidence in how to*

**Table 1** Interviewee demographics

| ID | Sex | Age (years) | Date interviewed | Household members |
|---|---|---|---|---|
| 1 | F | 61–70 | 08 June 2020 | Lives with spouse and teenage children |
| 2 | F | 61–70 | 11 June 2020 | Lives with husband with cancer |
| 3 | F | 41–60 | 12 June 2020 | Lives with teenage children |
| 4 | F | 61–70 | 29 June 2020 | Lives alone |
| 5 | F | 41–60 | 01 July 2020 | Lives with older parents with comorbidities, spouse and teenage child |
| 6 | F | 61–70 | 03 July 2020 | Lives with partner |
| 7 | F | 41–60 | 07 July 2020 | Lives with spouse and adult son |
| 8 | F | 41–60 | 16 July 2020 | Lives alone |
| 9 | M | 18–25 | 23 July 2020 | Lives with parents and sister |
| 10 | M | 26–40 | 10 September 2020 | Lives with partner |
| 11 | F | 61–70 | 21 September 2020 | Lives with husband with comorbidities |
| 12 | F | 26–40 | 28 September 2020 | Lives with partner |
| 13 | F | 26–40 | 05 November 2020 | Lives with partner |

*perform the behaviours.* The survey findings are discussed alongside the interview data within the themes which they mapped onto. Extracts from the survey data are delineated by the letter 's'.

## Perceived risk

Germ Defence encourages users to evaluate their own level of risk and which actions they feel are appropriate for them based on this level of risk, to enable users to focus on the behaviours and advice they deem the most personally relevant. For more detail on the intervention content and how we tailored it for perceived risk, see other publications from the project.[2] Participants' assessments of their level of risk played a major role in their willingness to engage in the protective behaviours, particularly those seen as more 'extreme' such as social distancing from other household members. Those who perceived that the virus is likely to enter their home, and/ or that household members are at risk of becoming seriously unwell were generally highly motivated to engage with the behaviours.

### Current levels of virus in circulation

Information about the current actual risk of infection was important for some people to help make decisions about performing difficult behaviours. For example, a mother justified her reluctance to follow social distancing guidance in the home in terms of the lower perceived necessity to do this at the moment.

*There is that sort of hope that, as there is I think known to be that much less of the virus out there generally at the moment… although we're still taking all the precautions, there is that hopefulness that the risk is less now than it was back in March.* (int 3)

*I didn't follow the stricter suggestions such as using disinfectant in the home, as we're low risk and the area we live in has very low numbers of cases.* (s71)

### Perceived likelihood of virus entering the home

Some participants were concerned about those in the household bringing the virus home if they needed to leave for work. This was influenced by how much mixing the person was doing outside the home, and the perceived severity of the consequences if someone in the household became ill.

*They said only one person is allowed out during the lockdown. So it was my husband… I was worried, because I'm the one who does the cooking and things, that I would pass it on to my parents if he caught it.* (int 5)

Having people from outside the household in the home was felt to be a significant risk. Participants were generally highly motivated to engage in the protective behaviours when visitors were present.

*I had a workman come in and he had to look at – because my heating's gone – and I was having a heart attack with him touching anything. So I was going round spraying everything with bleach like a maniac, even the carpet. So what are you meant to do if you've got workmen. I made him wear a mask, I made him wear gloves.* (int 5)

*Well I'm not going in anybody's house, and I'm not having anybody in my house…My house is my safe haven.* (int 4)

### Perceived risk of severe consequences to health

People's perceived risk of severe illness or death from the virus was influenced by comorbidities (such as cancer, chronic obstructive pulmonary disease, asthma and high blood pressure), old age, ethnicity and being an intergenerational household.

| Table 2 Survey respondent demographics | | |
|---|---|---|
| | **N** | **%** |
| Age (years) | | |
| 26–40 | 2 | 1.6 |
| 41–60 | 37 | 29.8 |
| 61–70 | 41 | 33.1 |
| 70+ | 31 | 25 |
| Missing | 13 | 10.5 |
| Experience with COVID-19 | | |
| I am at increased risk | 50 | 40.3 |
| Someone I live with is at increased risk | 19 | 15.3 |
| I think I've had COVID-19 | 7 | 5.6 |
| I think someone I live with has had COVID-19 | 1 | 0.8 |
| None of the above/no experience | 33 | 26.6 |
| Missing | 14 | 11.3 |
| Ethnicity | | |
| White British | 101 | 81.5 |
| White Irish | 1 | 0.8 |
| White European | 2 | 1.6 |
| White Canadian | 2 | 1.6 |
| Black British | 1 | 0.8 |
| Black African | 1 | 0.8 |
| British Chinese | 1 | 0.8 |
| Missing | 15 | 12.1 |
| Education level | | |
| Presecondary school | 1 | 0.8 |
| Secondary School | 43 | 34.7 |
| Undergraduate | 38 | 30.6 |
| Postgraduate | 28 | 22.6 |
| Missing | 4 | 11.3 |

*When you've taken a decision to tell your parents to come and live with you, and then you're reading stuff about intergenerational households, it's a much higher risk…* (int 5)

One participant described how she decided to shield with her husband to protect him, despite not being classed as vulnerable herself.

*I would just be so petrified I was going to give him something… I feel less… kind of imprisoned in a way, by shielding myself with him, than going out into the so-called freedom, but then coming back and being petrified I'll kill him.* (int 2)

Another described how one of the younger members of her household felt he didn't need to worry about the virus because of his age, and he perceived that only those at increased risk needed to be concerned.

*Our young man thinks that the only people that you should be worried about are people that are at increased risk, should*

*they catch it. Not everybody else. Do you know what I mean, it's like, oh well, it doesn't matter because they're fine, my friends are fine.* (int 1)

### Belief in the effectiveness of the protective behaviours

The perceived effectiveness of behaviours appeared to influence participants' willingness to engage with them. Participants identified an important caveat: the virus could spread within the home before symptoms present, meaning that protective behaviours could be viewed as pointless unless performed consistently. However, perceiving viral load to be a factor in viral transmission seemed to mitigate this, and these participants felt empowered to enact small changes around their home to reduce their risk.

#### Perceived value of cleaning

Most participants were already very aware of cleaning and washing hands and felt these were important. However, cleaning was sometimes associated with being paranoid and fearful, and some participants were keen to explain they weren't paranoid about the level of cleaning they do, while others described how the virus has made them feel paranoid about cleaning.

*Careful but not paranoid, yeah. I don't wash my keys in soapy water, and I don't regularly wash my car. We just wash and hand gel our hands after we've been somewhere that's in the car, when we get back into it.* (int 7)

*t the beginning I was cleaning constantly. I still am…. And then I'm spraying down the surfaces with disinfectant, because I'm worried about this transference. Okay, you've just touched it, so you've put it down. So that now gets onto that surface, if somebody in the meantime touches that surface, it then carries on and then goes onto another surface. That's what I'm on about, with the paranoia.* (int 5)

#### Perceived value of wearing a face covering

People's willingness to wear a face covering was strongly influenced by perceptions of effectiveness, although the focus was on wearing them outside the home. Most of the interviews took place prior to the mandatory use of face coverings in the UK, and there was some uncertainty and variance within the public discourse regarding their effectiveness at the time. These sentiments were reflected by our participants. Some people had read information from other countries which convinced them that face coverings were an effective way to prevent transmission, and one participant emphasised how she believed face coverings were important for protecting others more than yourself, whereas a few remained unconvinced and wanted more evidence.

*I might wear a mask, like I told you, I need to do more research on that.* (int 6)

Reasons offered for why masks might be ineffective included lack of filters, the mask causing infection due to dampness from breath, and people touching their face.

Furthermore, at the start of the pandemic and during the time in which most of the data collection took place, infection control strategies (including Germ Defence) placed a strong focus on surface transmission. As the pandemic progressed, the focus has shifted to airborne transmission, particularly the importance of ventilation. However, since manual transmission remains a potential transmission pathway within the home, Germ Defence was altered to additionally emphasise airborne transmission, rather than reduce the emphasis on handwashing and surface transmission. For more information on the advice given in Germ Defence and how this has changed during the progression of the pandemic based on Public Health England, PPI and stakeholder input, see further publications from the project.[16] This could explain why our participants reported stronger beliefs in the value of cleaning surfaces over face covering and ventilation.

### Barrier: Virus is likely to spread before you know you're ill

Some people were uncertain whether it would be achievable to prevent the virus spreading in the home.

*I think I probably still am, to a certain extent, sceptical about whether we would be able to get a virus come into this home and avoid spreading it between us.* (int 3)

People were concerned that the virus would already have spread by the time they socially distanced or self-isolated, making it pointless unless done continually.

*If at any stage I started to feel ill, which is probably then too late, because I probably would've then spread it to them, I could've potentially spread it to them by then anyway, I would then take myself to my room.* (int 3)

### Facilitator: Reducing all or nothing thinking

People were more likely to perceive protective behaviours as effective and worthwhile when they perceived catching the virus as a continuum based on how much viral load you are exposed to, rather than you either catch it or not.

*I use antibacterial wipes on just about all the shopping that comes into the house as well, when it's delivered, just as a precaution. Because I think it's safer if you do get the virus that it's as small as possible.* (int 7)

This was empowering as it helped people feel that small changes can still make a difference.

*I am sitting here thinking, if I turned the table the other way around, we could actually sit further apart from each other at the table, which might be one small thing.* (int 3)

Survey participants also highlighted the importance of balancing behaviours in accordance with personal risk level and perceived negative impact of the behaviour (eg, social distancing negatively impacting well-being), linking in with the perceived risk theme.

*It might not be good to be keeping them* (children) *at 2 m away for their development or mental health. Need more nuances about balancing risk against looking after child development.* (s18)

### Acceptability of distancing and isolation

Social distancing and isolation behaviours were presented on the Germ Defence website as recommended for higher-risk individuals, but also as useful ideas for lower-risk households to help reduce risk whenever it was deemed necessary. Spending time together was perceived as integral to the well-being of the household, but some participants described small changes they had made to help maintain intimacy while socially distancing or self-isolating. Social distancing and self-isolation were seen by some as only acceptable for short periods of time when symptoms were present.

### Barrier: Importance of time together

The idea of self-isolating within the home was quite daunting for people and there was some concern about the effect on mental well-being. Experiences of intimacy with partners and family members was generally judged to be of higher importance than reducing the risk of virus transmission when no symptoms were present, even when some members of the household were at high risk.

*I don't think I could cut down on the amount of time I spend with other people, because they'll get lonely…* (int 5)

*Because to a ninety-five-year-old a kiss is more important than worrying about whether or not you're going to die of a virus.* (int 11)

Some people described spending some time on their own during the day, but the evening meal was often regarded as an important time to spend together.

*The evening meals are nice… that's the one thing where we don't really take any precaution with the family, just because we all sit around the dinner table. But that is a nice part of the day, really, so in that respect it's quite good for everyone's mental health.* (int 9)

One couple found the idea of eating separately with the at-risk individual in his room as completely unacceptable:

*I think the guidance said something awful, like he should stay in his own room and be, you know, deliver his food to him like he was a kind of caged animal.* (int 2)

Some people perceived social distancing as acceptable for short periods of time if someone is ill, but not as something to do indefinitely as a preventative measure.

*Is that something I would have to do all the time, every day of my life? And then that feels completely… I wouldn't feel that there was much quality of life if I had to… if I'm living in the same house as my children at the moment but I couldn't hug them or sit near them or… It's something I could see potentially doing if it was for a limited period, but it just feels impossible sort of long-term.* (int 3)

### Facilitator: Ways of maintaining (distanced) intimacy

Some participants had made changes at home to enable social distancing, and they described how they managed to maintain some feelings of intimacy. Small changes to furniture arrangements or daily routines, the use of technology, and contact which was perceived to be low risk were seen as effective ways to engage with the protective behaviours without completely sacrificing intimacy and connectedness:

*I added on an extra table in the dining room, so that I could keep a metre from him when we're eating, even though it's joined eating.* (int 5)

*We have a bit of a dry cuddle, like I go over his shoulders, but I don't breathe on him and he doesn't breathe on me. So we're kind of on board with it, you know?* (int 2)

*In the morning, I go and wake him up and say, 'Oh, I'm getting up now for work,' and he goes down and makes me a cup of tea, just because we kind of like to have that… But he will deliver it to my dressing table and then I'll pick it up and take it back to bed. It's kind of trying to keep that intimacy, but without actually sharing everything.* (int 2)

### Having capacity to perform protective behaviours

This theme explores participants' perceptions of the practical factors which affect their capacity to perform the suggested behaviours. Having sufficient space was an important factor in how feasible it was for people to socially distance and self-isolate. Those who lived in smaller spaces generally found the idea of social distancing unfeasible.

*I'm guessing this applies to people in like houses more than just like one room, 'cause I currently live in a flat, a one bed flat with my partner, so it's kind of impossible for us to have one room in our home to be just for us.* (int 12)

Some people found it challenging trying to implement house rules for others to follow during the pandemic. Handwashing was a particular behaviour mentioned that participants tried to persuade partners and children to do, or checked whether they had done, which was identified as a source of tension.

*But when he comes home, I tell him to wash his hands, and every time he gets home, I'm always, 'Have you washed your hands?'* (int 6)

*I will just keep reminding him, all the time, to wash his hands. And he'll say, 'I've done it.' You say, 'No you haven't. The sink's not wet.' And, "well I did it. I did do it, I did it when I got to my…' Because he's a sink in his room, 'I did it when I got to my room' which we know is not necessarily the case. So it's… it's tricky, but we're trying to keep on the case.* (int 1)

### Habit forming reduces effort

When discussing the effort involved in performing the protective behaviours, participants typically discussed how well they integrated with their current behaviours and routines. Some participants described how some protective behaviours, such as cleaning, regular handwashing and not sharing towels, had already been the norm for them before the pandemic, which helped them to adhere.

*I found that they are things that I have always done, throughout my life, because I was taught to as a child.* (int 4)

Social distancing was also facilitated in some households with teenage children, who were described as spending a lot of time in their rooms anyway. Additionally, despite an initial negative reaction to social distancing, some described how working from home meant that they were spending most of their time away from other household members.

*I'm looking at it going, 'Really? You think this is a rational thing to do?' Like I mean, I do sit in a room on my own for most of the day, funnily, because I work from home, as does my partner, and you know, it… she'd irritate me if she was on calls and vice versa, so yeah, we do sit separately.* (int 10)

Where new behaviours had become habits for our participants, they perceived less effort involved in performing the behaviours.

*I think they're definitely becoming habits now. I mean, it is… still is harder than it used to be, because I never would've done that before. But it is more normal now.* (int 3)

Others who were being extremely careful about cleaning found it could be quite effortful and fatiguing. It seemed that participants living with people at increased risk were more likely to find the constant cleaning demanding.

*It feels like it's a constant state of vigilance. It's very high intensity, that level of concentration all the time, not to lapse.* (int 5)

### Confidence in how to perform the behaviours

Both interview and survey participants identified that they wanted clear and consistent practical information on what to do. Inconsistent information seemed to undermine people's confidence in their ability to perform the behaviours and reduce their risk.

*It's easier now than when it first started…I feel like the mask guidance just came out of nowhere, so 1 min they're telling us that they don't have any scientific evidence, and the next minute it's, 'from the 30th you have to wear masks,'… it was just strange…* (int 12)

However, both interview and survey participants felt more confident in their ability to engage with and perform the behaviours when they felt well informed and affirmed by those who they perceived to be experts.

*Knowing the advice came from trusted source gave me confidence and so helped to avoid fear/anxiety overwhelming.* (s81)

When participants felt that they were doing the 'right' thing, they felt empowered and motivated to continue. The Germ Defence website encourages users to plan how much they intend to engage with the behaviours going forward. If their plans show that their adherence will improve, they are given positive reinforcement:

*It's quite validating…I've reconsidered what I've been doing and now I'm going to make the steps, and I feel quite empowered.* (int 12)

### Social norms affect motivation to engage in the behaviours

Some participants discussed how they felt demotivated to engage in the behaviours when they perceived others were disregarding infection control advice. These participants felt that protective behaviours were simply 'not worth the effort' when others were not playing their part.

*I feel a little bit disenchanted by the whole thing, because you know, I've done things properly, … I didn't leave the house for… 3 months. And even when it was relaxed I didn't, and yet I still have to watch my neighbour, who's seventy-five, going out for a drive every single day during lockdown, and that is difficult to take. So it was a bit like 'actually … why am I doing my bit here, when everyone else isn't?'* (int 10)

### DISCUSSION

These findings show how people conceptualise the risk of catching and transmitting COVID-19, and use this as a rationale for their behaviour at home. In line with PMT[12] and a previous review of beliefs influencing protective behaviours during the swine influenza pandemic,[13] perceived risk of the virus and perceived effectiveness of the protective behaviours increased willingness to adhere. Cleaning and handwashing were widely perceived to be effective and acceptable, although some participants described how other members of their household were less adherent to these behaviours which could cause anxiety. Participants also found the protective behaviours easier if they fit well with their usual routine, suggesting that linking the new behaviours to more ingrained habits could increase adherence.

Behaviours such as spending time in separate rooms at home and keeping 2 m apart were less acceptable, especially as preventative measures to follow even when no one in the household has any symptoms (although this was only suggested for higher-risk households). Our participants generally felt that a lack of physical and emotional closeness with their household members was too much of a sacrifice to engage in social distancing regularly, even when the household was identified as high risk. Additionally, since our participants tended to find the behaviours easiest to adhere to when they fit well with their usual routine and when they formed a habit, it could be that these particular behaviours are seen as too different from their typical way of life. Finally, awareness of the concept of viral load helped people feel more

empowered as they understood that even small changes, such as spending *some* time apart, were worthwhile. This finding is consistent with the importance of *attitudes* and *perceived behavioural control* from the TPB.

There was some congruency between the current findings and previous research into adherence to infection control behaviours during a pandemic. The concern about being perceived as paranoid (within the subtheme; *perceived value of cleaning*) indicated that there may still be negative social connotations surrounding hygiene practices,[15] and supports the relevance of *social norms* from the TPB.[10] Further, our findings regarding the need for emotional connection and intimacy provide support for recent qualitative research into the impact of COVID-19 and adherence to government guidance, which showed that some may only partially adhere to the behaviours due to the need for and cultural importance of social contact, and some reported feelings of loss and grief over the loss of social interaction during lockdown.[21–23] Concerns about the negative impact of self-isolation, both in terms of practical logistics and emotional well-being, were also raised in a qualitative study with people who had been in contact with someone with COVID-19.[24] This suggests that self-isolation is a very difficult behaviour for many people even when risk is known to be high, and that appropriate support is essential. Additionally, the need for clarity and consistency in government and public health guidance has also been highlighted in other studies as important in aiding the public to adhere to infection control behaviours.[21 23 25]

Finally, our participants expressed some concern and awareness that transmission to other household members may well have occurred by the time that symptoms present, supporting previous qualitative research into the public's opinions of the need for separate accommodation for at-risk individuals during the COVID-19 pandemic.[26] This indicates a need for preventative educational interventions so that the public are equipped to act as soon as they feasibly can.

### Strengths and limitations

Triangulation of the think-aloud data with open-ended survey data revealed very high affinity between the two data sets, suggesting that the themes identified are valid and robust. However, the transferability of our results should still be treated with some caution due to the rapidly changing nature of the pandemic and government guidance, and because our sample may not represent the views of the general population. Half of our interview participants were Germ Defence users, recruited after receiving the intervention. It is therefore likely that they were more engaged and motivated than the general population since they sought out the intervention for themselves and subsequently volunteered to participate in research. The interview data gathered from non-website users did not differ substantially from the website users' experiences, although these volunteers are also likely to have an above average interest in reducing

transmission. Similarly, our survey had a low response rate of only 23%, suggesting that the findings may not be representative of the barriers to protective behaviours experienced by the wider population.

Our survey sample was also predominantly white British, and no interview participants identified themselves as belonging to Black, Asian and ethnic minority groups. While efforts were made to purposively sample for greater diversity, the need for rapid data collection to inform the optimisation of the intervention limited our recruitment options. However, as noted above, some similar concepts to the current findings were found in a recent interview study which focused on members of low-income and ethnic minority households.[22]

Our qualitative interviews were conducted via telephone due to the pandemic, but this remote method of data collection did not seem to negatively influence the richness or quality of the data. Participants appeared happy to share in-depth stories about their experiences and perceptions of the behaviours, and this is consistent with other research which has supported the value of remote qualitative research.[27]

## Conclusions and implications

Our findings have several implications for behavioural interventions and public health guidelines during a pandemic. These findings have shown that the public may be unwilling to adhere to the protective behaviours indefinitely if they perceive the risk to be low, so it is important that behavioural guidelines encourage accurate perceptions of personal risk level and highlight that enacting even small changes would still be worthwhile for reducing risk. People understood the concept of viral load and found this a helpful rationale for making small changes which could be maintained over time. Furthermore, the perceived negative impact of social distancing and isolation on mental well-being within the home seems to be a major sticking point in terms of the public's willingness to adhere. Behavioural interventions which offer practical suggestions for how intimacy could be maintained while socially distancing could reassure the public that they could reduce the negative impact on their well-being while engaging with protective behaviours, at least some of the time.

## Author affiliations

[1]School of Psychology, University of Southampton, Southampton, UK
[2]Department of Psychology, University of Bath, Bath, UK
[3]NIHR Biomedical Research Centre, Faculty of Medicine, University of Southampton, Southampton, UK
[4]The Quality Safety and Outcomes Policy Research Unit, University of Kent, Canterbury, UK
[5]Primary Care Research Centre, University of Southampton, Southampton, UK
[6]School of Psychological Science, University of Bristol, Bristol, UK

**Acknowledgements** The authors thank their voluntary research assistants, Amina Khan and Lara Rosa, who led engagement with social media and charities for study recruitment.

**Contributors** KM: conceptualisation, formal analysis, investigation, methodology, project administration, data curation, resources, validation, writing—original

draft preparation, writing—review and editing. LT: guarantor, conceptualisation, formal analysis, investigation, methodology, project administration, data curation, resources, validation, writing—original draft preparation, writing—review and editing. JG: investigation, project administration, writing—review and editing. SM: conceptualisation, resources, writing—review and editing. BA: conceptualisation, funding acquisition, resources, writing—review and editing. JD: software, resources, writing—review and editing. CR: resources, writing—review and editing. JB: resources, writing—review and editing. MW: resources, writing—review and editing. PL: conceptualisation, funding acquisition, resources, writing—review and editing. LY: conceptualisation, funding acquisition, resources, writing—review and editing.

**Funding** The study was funded by United Kingdom Research and Innovation Medical Research Council (UKRI MRC, https://mrc.ukri.org/) Rapid Response Call: UKRI CV220-009. The Germ Defence intervention was hosted by the Lifeguide Team, supported by the NIHR Biomedical Research Centre, University of Southampton. LY is a National Institute for Health Research (NIHR) Senior Investigator and theme lead for University of Southampton Biomedical Research Centre. LY is affiliated to the National Institute for Health Research Health Protection Research Unit (NIHR HPRU) in Behavioural Science and Evaluation of Interventions at the University of Bristol in partnership with Public Health England (PHE). MW is an NIHR Academic Clinical Lecturer, under grant CL-2016-26-005.The views expressed are those of the author(s) and not necessarily those of the NHS, the NIHR, the Department of Health or PHE. The funders had no role in study design, data collection and analysis, decision to publish, or preparation of the manuscript.

**Competing interests** None declared.

**Patient consent for publication** Not applicable.

**Ethics approval** This study involves human participants and was approved by the University of SouthamptonPsychology Ethics Committee (ID: 56445). Participants gave informed consent to participate in the study before taking part.

**Provenance and peer review** Not commissioned; externally peer reviewed.

**Data availability statement** All data relevant to the study are included in the article or uploaded as supplementary information. Full interview transcripts and survey responses cannot be shared publicly because our participants only consented to extracts being made available to those outside of the study research team.

**ORCID iDs**
Katherine Morton http://orcid.org/0000-0002-6674-0314
Lauren Towler http://orcid.org/0000-0002-6597-0927
Sascha Miller http://orcid.org/0000-0002-1949-5774
Ben Ainsworth http://orcid.org/0000-0002-5098-1092
James Denison-Day http://orcid.org/0000-0003-0223-0005
Cathy Rice http://orcid.org/0000-0001-5961-2413
Jennifer Bostock http://orcid.org/0000-0001-9261-9350
Merlin Willcox http://orcid.org/0000-0002-5227-3444
Paul Little http://orcid.org/0000-0003-3664-1873
Lucy Yardley http://orcid.org/0000-0002-3853-883X

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
