## [Reviewer comments · BMJ Open]

ARTICLE DETAILS

TITLE (PROVISIONAL)	Infection control in the home: A qualitative study exploring perceptions and experiences of adhering to protective behaviours in the home during the COVID-19 pandemic
AUTHORS	Morton, Katherine; Towler, Lauren; Groot, Julia; Miller, Sascha; Ainsworth, Ben; Denison-Day, James; Rice, Cathy; Bostock, Jennifer; Willcox, Merlin; Little, Paul; Yardley, Lucy

VERSION 1 – REVIEW

REVIEWER	Ohta, Ryuichi Unnan City Hospital
REVIEW RETURNED	15-Aug-2021

GENERAL COMMENTS	Thank you for giving me to review your manuscript. This manuscript is interesting and scientifically meaningful for considering healthcare behaviors in the COVID-19 pandemic. However, the following points should be revised for validity and reliability. 1. In the objectives of the abstract, the authors should describe what Germ Defense is and adherence to what.2. In the abstract results, the authors should describe participants' backgrounds for the transferability of this research.3. In the abstract results, the authors should describe the backgrounds of 124 Germ Defense users.4. In the introduction, the authors stated that adherence to infection control methods is mixed. What do they mean by "mixed"? They should clarify this part for clarity.5. In the introduction, the author should describe general issues of infection control methods before the description of Germ Defence. After that, the specialty of Germ Defence should be described.6. What is the theoretical framework? What is the research question of this research? The authors should describe them clearly for the credibility of this research.7. In the section on participants, the authors should describe how to collect the participants in specific numbers. For example, why did this study collect seven from the website and six from organizations and community groups? This is a critical point in this study.
---

	8. The authors should describe how to analyze the data from the participants from different categories in this qualitative study. 9. This study used quantitative data, so this study's design should be a mixed-method approach. 10. This study's method section is difficult to read. The contents were mixed. The authors should allocate the appropriate contents to proper sections. For example, the participants' backgrounds should be located in the result section. 11. In the thematic analysis, data quality is essential. This research's weak point is why the researchers analyzed the data from different category's participants in the same analysis. This is a critical point for the trustfulness of this research. 12. In the analysis section, the authors should describe the theoretical saturation and member checking clearly for the research credibility. The web-based survey had a low collection rate, which can be weak for triangulation for this research. 13. The discussion part should summarize this research's results and significant points by revising the previous issues.
--	---

REVIEWER	Bozdag, Faruk Istanbul Universitesi-Cerrahpasa
REVIEW RETURNED	29-Aug-2021

GENERAL COMMENTS	Dear authors, First of all, I would like to thank you for this important study you have provided to the literature. I have carefully examined your article. My evaluation and recommendations about the manuscript are follow. With kind regards. It should be stated which of the qualitative approach is used. The discussion section should be enriched with relevant research results. Also discuss the theoretical consistency of the current findings with TPB and PMT. Please indicate the limitations of the think-aloud interviews via phone in determining the perceptions and experiences of the participants
--

VERSION 1 – AUTHOR RESPONSE

Reviewer: 1

Dr. Ryuichi Ohta, Unnan City Hospital

Thank you for giving me to review your manuscript. This manuscript is interesting and scientifically meaningful for considering healthcare behaviors in the COVID-19 pandemic. However, the following points should be revised for validity and reliability.

Thank you for your positive comments, and for your helpful feedback. We have addressed the points you raised below.

1. *In the objectives of the abstract, the authors should describe what Germ Defence is and adherence to what.*

We have added more detail about the Germ Defence intervention under the abstract heading 'intervention' (page 2), which was also in line with the Editor's feedback. We would like to include more information about its development and theoretical basis, but due to the tight word count in the abstract we felt it was more important to use the available words to explain our methods in more detail.

We reference a number of other publications in the Introduction that have discussed the Germ Defence intervention in more detail, and we have focused on the relevant aspects of its developmental and theoretical basis for the current paper.

2. *In the abstract results, the authors should describe participants' backgrounds for the transferability of this research.*

We agree this was an important omission and we have now added details of the participants' demographics to the 'Participants' section of the abstract (page 2):

"Participants: Thirteen people took part in telephone interviews, and 124 completed a qualitative open-text survey. All were recruited from the public. Most survey participants were aged over 60, while interview participants were more distributed in age. Most reported being at increased risk from COVID-19, and White British"

3. *In the abstract results, the authors should describe the backgrounds of 124 Germ Defence users.*

We have now included more detail of the 124 survey participants' demographics in the Participants section of the abstract (page 2):

"Most survey participants were aged over 60, while interview participants were more widely distributed in age. Most reported being at increased risk from COVID-19, and White British"

4. *In the introduction, the authors stated that adherence to infection control methods is mixed. What do they mean by "mixed"? They should clarify this part for clarity.*

We have now replaced the word 'mixed' with 'varies widely' which we think is clearer (page 4):

"However, evidence suggests that adherence to these behaviours varies widely in the UK and other affected countries, suggesting there may be challenges for people in implementing these behaviours in a real life setting.[2-4]"

5. *In the introduction, the author should describe general issues of infection control methods before the description of Germ Defence. After that, the specialty of Germ Defence should be described.*

We have now expanded para 1 of the Introduction to further detail the infection control behaviours and the rationale for understanding these (page 4):

“Behavioural measures have been recommended to help control the spread of the COVID-19 virus, including hand-washing, cleaning surfaces, mask-wearing, social isolation, and social distancing.[1] However, evidence suggests that adherence to these behaviours varies widely in the UK and other affected countries, suggesting there may be challenges for people in implementing these behaviours in a real life setting.[2-4] Transmission of COVID-19 within the home is a key risk,[5,6] therefore understanding barriers to adhering to protective behaviours within the home could be particularly important”.

6. *What is the theoretical framework? What is the research question of this research? The authors should describe them clearly for the credibility of this research.*

Para 2 of the Introduction (Page 4) outlines the theories that informed the development of the behavioural intervention; specifically TPB, common sense model and PMT. We did not use a theoretical framework for the study as we wanted to inductively explore the barriers and facilitators that were most relevant to people.

The research question is detailed in Para 3 of the Introduction (Page 5):

“This study sought to explore experiences and perceptions of performing protective behaviours at home in order to identify possible barriers and facilitators, and develop an understanding of how these behaviours are influenced by perceptions.”

7. *In the section on participants, the authors should describe how to collect the participants in specific numbers. For example, why did this study collect seven from the website and six from organizations and community groups? This is a critical point in this study.*

Thank you for pointing this out, we have now explained in the participants section of the method why these numbers were sampled (Pages 5-6):

“Inclusion criteria were those over the age of 18, able to access the Germ Defence website and able to give informed consent. Users of the Germ Defence website were invited to register their interest in taking part in research to optimise the website. Seven interview participants were purposively sampled from the volunteers by factors such as age, gender, education level, risk status and experience of COVID-19 to maximise diversity.

However, after seven interviews we identified that these participants were mostly highly educated about infection control behaviours and highly motivated to adhere. As we wanted to understand barriers amongst people with lower levels of awareness and motivation, we recruited the remaining participants via social media and newsletters sent out by organisations and community groups to target people who had not already sought out the Germ Defence intervention (n=6). We stopped recruiting once we felt we had reached saturation and that no new barriers or facilitators were being identified”.

8. *The authors should describe how to analyze the data from the participants from different categories in this qualitative study.*

We now clarify in the Data Analysis that the data from the interview participants and the data from the qualitative survey were analysed completely separately and inductively, and only subsequently were the findings of each thematic analysis triangulated (page 12):

“Responses to the four open-text survey questions were coded inductively using thematic analysis, separately from the interview data analysis”.

9. *This study used quantitative data, so this study's design should be a mixed-method approach.*

No quantitative data were included in this paper, so it was an entirely qualitative paper. We have now explained in the abstract and initial strengths and limitations that the survey was a *qualitative* survey, to avoid causing confusion to readers (pages 2 and 3)

10. *This study's method section is difficult to read. The contents were mixed. The authors should allocate the appropriate contents to proper sections. For example, the participants' backgrounds should be located in the result section.*

We apologise for the confusing structure of the methods. We have now moved the demographics tables to the Results (page 13-15), and re-ordered them to follow the consistent structure of reporting the interview data first and then the qualitative survey data.

We have also added subheadings to clarify which paras relate to the interviews and which relate to the survey.

11. *In the thematic analysis, data quality is essential. This research's weak point is why the researchers analyzed the data from different category's participants in the same analysis. This is a critical point for the trustfulness of this research.*

We believe that the independent inductive analysis of the qualitative interviews and qualitative survey, followed by subsequent triangulation of the findings, is a strength of the paper rather than a weakness. This is because using multiple methods to explore perceptions can generate a more in-depth understanding and enable the inclusion of the views of more diverse participants (Johnson et al 2017 <https://bmcmmedresmethodol.biomedcentral.com/articles/10.1186/s12874-017-0290-z>).

Our data analysis section now clarifies that the qualitative survey analysis was conducted completely independently of the interview thematic analysis, and how our approach to triangulation influenced this decision (page 12-13):

“Responses to the four open-text survey questions were coded inductively using thematic analysis, separately from the interview data analysis. The resulting categories were then mapped onto the themes generated from the interview data to assess their fit with these themes, whether any new themes or subthemes were present in the survey data, and to what extent the survey data provided further nuance to the existing themes”.

The finding that the qualitative survey data findings mapped well onto the themes generated by the interviews was encouraging in terms of the robustness of the findings.

12. *In the analysis section, the authors should describe the theoretical saturation and member checking clearly for the research credibility. The web-based survey had a low collection rate, which can be weak for triangulation for this research.*

We agree that it's important to mention saturation, member checking and the limitation of the low response rate to the qualitative survey, and apologise for these omissions.

Details about saturation were added to the participants section of the methods (page 6):

“We stopped recruiting once we felt we had reached saturation and that no new barriers or facilitators were being identified”.

The data analysis section now explains that findings were checked with participants via a newsletter (page 12):

“Findings were shared with participants via a newsletter, and participants were invited to contact the research team if they had any feedback on the findings”.

The strengths and limitations at the start of the paper and the discussion section now mention the issue of the low survey response rate:

“In addition, the qualitative survey had a low response rate which could limit transferability”.
(page 4)

“Similarly, our survey had a low response rate of only 23%, suggesting that the findings may not be representative of the barriers to protective behaviours experienced by the wider population,” (page 30)

13. *The discussion part should summarize this research's results and significant points by revising the previous issues.*

We have added the additional limitations to the discussion (page 29-30).

Reviewer: 2

Dr. Faruk Bozdog, Istanbul Universitesi-Cerrahpasa

First of all, I would like to thank you for this important study you have provided to the literature. I have carefully examined your article. My evaluation and recommendations about the manuscript are follow.

Thank you for your positive comment. We have addressed the points you raised below.

It should be stated which of the qualitative approach is used.

We apologise for not being clear. We now clarify that we used inductive thematic analysis (Page 12).

The discussion section should be enriched with relevant research results. Also discuss the theoretical consistency of the current findings with TPB and PMT.

Thank you we have now added to the discussion to show how the findings fit with TPB and PMT, as well as linking with further recent research findings to those that were already mentioned in Paras 3 and 4 of the discussion (Pages 27-29):

“In line with Protection Motivation Theory[12] and a previous review of beliefs influencing protective behaviours during the swine flu pandemic,[13] perceived risk of the virus and perceived effectiveness of the protective behaviours increased willingness to adhere”

“This finding is consistent with the importance of *attitudes* and *perceived behavioural control* from the Theory of Planned Behaviour.”

“Concerns about the negative impact of self-isolation, both in terms of practical logistics and emotional well-being, were also raised in a qualitative study with people who had been in contact with someone with COVID-19 (ref).[25] This suggests that self-isolation is a very difficult behaviour for many people even when risk is known to be high, and that appropriate support is essential”.

Please indicate the limitations of the think-aloud interviews via phone in determining the perceptions and experiences of the participants.

Thank you for raising this important point, we have now added a reflection on the limitations of telephone interviews to the discussion (page 30):

“Our qualitative interviews were conducted via telephone due to the pandemic, but this remote method of data collection did not seem to negatively influence the richness or quality of the data. Participants appeared happy to share in-depth stories about their experiences and perceptions of the behaviours, and this is consistent with other research which has supported the value of remote qualitative research.[28]”

VERSION 2 – REVIEW

REVIEWER	Ohta, Ryuichi Unnan City Hospital
REVIEW RETURNED	08-Nov-2021
GENERAL COMMENTS	The manuscript has been considerably improved. I think that this paper is suited for inclusion in our journal.